# DSML: Dual Semi-Supervised Meta-Learning for Multi-Lingual Text Classification

## Abstract

Multi-lingual text classification (MLTC) is a challenging task since it faces language differences and scarce annotated data (especially in low-resource language). To improve the model's multilingual comprehension and text classification ability with a small amount of annotated data, this paper proposes a dual semi-supervised meta-learning method (DSML). Specifically, DSML constructs a teacher-student framework and uses dual meta-learning to help the teacher and student collaborative evolution (co-evolution). The teacher and student models are both initialized with text classification ability with a limit annotated data. The teacher model is also initialized with better multi-lingual comprehension ability than the student model. Through a co-evolution mechanism, the student model granularity learns the teacher model's multilingual comprehension ability, ultimately improving both multi-lingual comprehensive ability and text classification ability. We conduct extensive experiments on a newly collected MLTC dataset and the experiments show that our DSML model achieves a state-of-the-art performance. We give a detailed analysis of the experiments and the data/code will be released on GitHub.

## 1 Introduction

Multi-lingual text classification (MLTC) tasks are widely used in practical scenarios such as public opinion analysis and industry intelligence analysis (Brauwers & Frasincar, 2023; Minaee et al., 2022). The divergent nature of languages and cultures poses a significant challenge to applying a uniform set of classification criteria across multilingual text data. This process requires annotators to harmonize not only linguistic distinctions but also variations in annotation standards, rendering the task particularly difficult (Snell et al., 2017; Bao et al., 2020). Hence, how to improve classification models' performance with a limited number of annotated samples became current research hotpoints (Liu et al., 2023; Shliazhko et al., 2024).

Semi-supervised learning (SSL) is one of the widely used techniques in text classification when labelled training data is limited. SSL annotated unlabelled samples by using a teacher classifier trained on both labelled data and previously pseudo-labelled data, and repeatedly iterates this process in a self-training loop (Tanantong & Parnkow, 2022). For example, Mean Teacher (Tarvainen & Valpola, 2017a) maintained a teacher model and used the exponentially weighted average of the student model parameters to generate stable pseudo-label targets for unlabelled data, thereby enhancing the model's robustness. MultiMatch (Sirbu et al., 2025) selected and filtered pseudo-labels based on the consistency of head part differences and model confidence, and allocates weights based on the perceived classification difficulty. However, these studies mainly focus on monolingual scenarios and do not consider language differences in the training data.

In multi-lingual scenarios, Ghosh et al. (2022) used GNN and Transformer to capture the linguistic feature of multiple languages and facilitate the MLTC task. Yunis (2024) addressed multi-language text classification tasks by simplifying complex problems into single-label classification and separately processing data of different languages. Gao et al. (2025) introduced external knowledge into meta-learning mechanism to learn better generality ability. However, the training objectives of multilingual understanding tasks differ from those of text classification, existing methods lack integration of different training objectives. The absence of explicit reinforcement of linguistic comprehension tasks during the training of multilingual text classification models can result in the degradation of

this critical capability, thereby hindering the enhancement of classification accuracy Gudibande et al. (2023) Jia & Liang (2017), despite linguistic comprehension being fundamental to the task.

To address the aforementioned challenges, this paper proposes a dual semi-supervised meta-learning learning (DSML) for MLTC tasks. DSML contains three steps: in step 1, we train the model's foundational capabilities using a small amount of labelled data; in step 2, we train the student model's multilingual classification ability using a multilingual pseudo-label corrector and the teacher model. In Step 3, we determine whether the student model has converged using a small amount of annotated data. If it has not converged, we use the student model to update the teacher model through back-propagation and return to step 2, so that the teacher model can generate higher quality of pseudo-labels to guide the training process.

Experimental results show that this method significantly improves the classification performance on Chinese, Vietnamese, and English data. Meanwhile, the multi-lingual understanding ability of the teacher model also improves. The advantages of DSML are: 1) it does not need external knowledge besides the labelled and unlabelled training data, making it suitable for MLTC tasks (especially low-resources language); 2) it helps to improve the abilities of both teacher and student models through collaborative evolution, so that the potential of the models in downstream tasks has been better explored. The contributions are as follows:

- We proposed a dual semi-supervised meta-learning learning (DSML) for MLTC tasks, which jointly prompts multilingual comprehension and text classification.
- We design a noise-aware semi-supervised learning strategy which uses a multilingual pseudo-label corrector to speed up the two-way learning process through pseudo-label optimization and feedback-driven update.
- Extensive experiments show that DSML achieves new state-of-the-art performance on MLTC tasks. We give a detailed analysis of the experimental results. The code and data of this paper will be released on GitHub.

## 2 RELATED WORK

The two core characteristics of the multilingual text classification task are language diversity and scarcity of annotation resources. Especially for low-resource languages, the cost of obtaining annotated data is extremely high. To address this issue, current mainstream methods achieve efficient cross-language transfer through semi-supervised learning and few-shot learning, significantly reducing the reliance on manual annotation.

### 2.1 SEMI-SUPERVISED LEARNING FOR MLTC

Semi-Supervised Learning (SSL) demonstrates great potential in many fields by simultaneously utilizing labelled and unlabelled data, such as in smart agriculture (Shorewala et al., 2021), autonomous driving (Lin et al., 2025), legal justice (Li et al., 2022), and energy management (Silva et al., 2011). Current SSL methods for text classification mainly focus on consistency regularization, pseudo-labeling, and hybrid augmentation techniques. Consistency regularization methods (Sirbu et al., 2025; Zhao & Yao, 2022) enforce prediction stability under perturbations, with MultiMatch (Sirbu et al., 2025) filtering pseudo-labels based on consistency and confidence, while Mean Teacher (Tarvainen & Valpola, 2017b) uses EMA predictions as supervision. These perform well monolingually but face challenges with multilingual data due to cross-lingual feature disparities. Pseudo-labeling techniques expand training data iteratively, where traditional approaches use fixed thresholds (Lee, 2013), while recent work improves robustness via dynamic thresholds (e.g., Curriculum Pseudo Labeling (Zhang et al., 2021a)) and noise-aware mechanisms (e.g., total variation regularization (Zhang et al., 2021b) and dual meta-learning (Li et al., 2024)). Hybrid methods like UDA (Xie et al., 2020) combine augmentations (e.g., back-translation) with consistency training, performing well in high-resource languages but remaining understudied in low-resource settings. Recent studies on large language models (LLMs) show that fully fine-tuned models achieve top performance in text classification (Zhao et al., 2024), while XLM-R fine-tuning reveals persistent challenges in low-resource language scenarios (Homskiy & Maloyan, 2023). Hatefi et al. (2025) (Hatefi et al., 2025) introduce an unsupervised pre-training phase based on objective masking, and conduct in-depth performance evaluations of the original model to improve the performance

of DML. **Current semi-supervised methods still inadequately exploit multilingual feature correlations and lack robust teacher-student collaboration. Our approach addresses these gaps through noise-aware SSL and a multilingual pseudo-label corrector, enhancing robustness via optimization and feedback mechanisms.**

### 2.2 META-LEARNING FOR MLTC

Meta-learning, which aims to enable models to extract generalizable knowledge from past learning experiences, has become a powerful framework for rapid adaptation to new tasks, particularly in data-scarce scenarios (Finn et al., 2017; Li et al., 2017; Nichol et al., 2018). As a "learning to learn" paradigm, it operates through an interaction mechanism where the meta-learner updates the base learner's parameters, maintaining a dynamic equilibrium between internal stability and external adaptability. Current meta-learning approaches include optimization-based methods like MAML (Finn et al., 2017), which learns optimal initialization parameters for fast adaptation, and metric-based techniques such as Prototypical Networks (Snell et al., 2017), effective in multilingual settings but sensitive to cross-lingual feature discrepancies. Recent prompt-based methods like PBML (Zhang et al., 2022) and EMPT (Lv, 2024) leverage natural language prompts and efficient classifiers to enhance few-shot learning. **However, existing methods often suffer from issues like multi-task optimization imbalance and insufficient collaboration between learning objectives. To address these challenges, this paper proposes a teacher-student collaborative evolution mechanism for multilingual text classification, using dual meta-learning to jointly improve multilingual understanding and classification performance.**

## 3 METHOD

### 3.1 PROBLEM DEFINITION

The training dataset $\mathcal{D}$ consists of a labelled subset $\mathcal{D}^l$ and an unlabelled subset $\mathcal{D}^u$. The number of samples in $\mathcal{D}^l/\mathcal{D}^u$ is $N/M$, respectively. $M>>N$ indicates that there are significantly more unlabelled data than labelled data. Semi-supervised learning assume that both $\mathcal{D}^l$ and $\mathcal{D}^u$ follow the same data distribution $P(x,y)$, where $x_i^l \in \mathcal{D}_l$ represents the $i$-th labelled data, $x_i^u \in \mathcal{D}_u$ represents the $i$-th unlabelled data, and $y_i^l$ represents the one-hot label vector of $x_i^l$. The goal of this paper is to learn a MLTC model that give a correct label for an input text.

### 3.2 OVERVIEW OF THE PROPOSED METHOD

Figure 1 show the DSML framework.

**In Step A**, we train the model's foundational capabilities using a small amount of labelled data. Specifically, the teacher/student models are trained with labelled MLTC data for text classification ability (as shown in Step A: 1-2-4 and 1-5-6 of Figure 1), The teacher model is also additionally trained using labelled data through a masked-language-modelling task to enhance multilingual comprehension ability. The loss function for this task is the standard MLM loss (as shown in Step A: 1-2-3 of Figure 1). We hope to gradually integrate teacher's multilingual comprehension ability with student's multilingual classification ability through subsequent collaborative training, in order to achieve stronger classification performance than training alone;

**In Step B**, we train the student model's multilingual classification ability using a multilingual pseudo-label corrector and the teacher model. Specifically, the teacher and student model generate labels for the unlabelled data (as shown in Step B: 1-3-5 and 1-4-6 of Figure 1). Then a multi-lingual pseudo-label corrector utilize a correction mechanism to correct labels from the student (as shown in Step B: 1-2). Then corrected label is trained to align with the label from the teacher (as shown in Step B: 7 of Figure 1). Then the teacher and student are trained with distillation loss with their logits (as shown in Step B: 8 of Figure 1). The student and the teacher is then updated with the corrected/hard pseudo labels,respectively (as shown in Step B: 3-5 and 4-6 of Figure 1). Finally, the labelled data is used to generate teacher's Logit (as shown in Step B: 9-10 of Figure 1) and is used to update the multilingual pseudo-label corrector (11 and 12). By recycling labelled data and unlabelled data, we hope to improve the ability of both models (multilingual pseudo-label corrector and the teacher model) to generate high-quality pseudo labels.

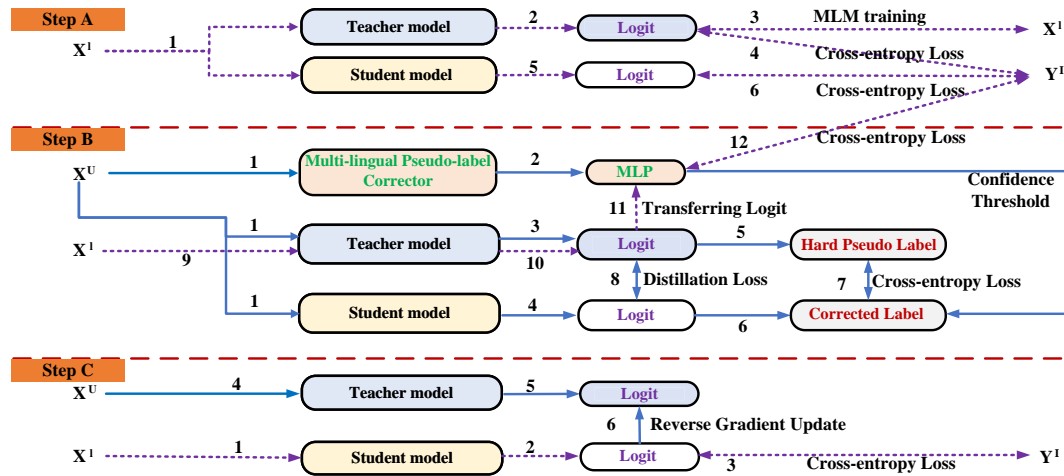

Figure 1: The proposed DSML method.

**In Step 3**, we determine whether the student model has converged using a small amount of annotated data (as shown in Step C: 1-2 of Figure 1). If it has not converged, the performance of the student classifier on the real-labelled data serves as a feedback signal (as shown in Step C: 4-5-6 of Figure 1), and through the meta-learning mechanism, the parameters of the teacher classifier are dynamically optimized and adjusted to make the pseudo-labels generated by the teacher classifier more conducive to the improvement of the generalization ability of the student model, forming a closed-loop system of collaborative evolution.

## 3.3 STEP A: FOUNDATIONAL MODEL TRAINING

Models fundamental capabilities are established through joint training and fine-tuning in Step A.

### 3.3.1 TEACHER MODEL

Based on the design goal of the co-evolution framework, we constructed a teacher model that focuses on optimizing multilingual capabilities, concentrating on extracting more discriminative language features. It consists of a text encoder, a language feature transformation layer (MLP), and a classifier. We used multilingual BERT (mBERT) as the basic text encoder to obtain the context representation of the input sequence. Specifically, for the input sequence $x_i$ in the training set, we obtain its context-dependent vector representation through mBERT: $\mathbf{H}^l = \mathrm{mBERT}(\texttt{[CLS]}\,x^l\,\texttt{[SEP]})$.

Here, $\mathbf{H}^l = [h_1^l, h_2^l, ..., h_{|H|}^l]$ represents the hidden output representation of the mBERT model, where $\texttt{[CLS]}$ and $\texttt{[SEP]}$ are special tokens. $h_i^l$ represents the $i$-th token in $\mathbf{H}_i$. The hidden state representation of the classification token $\texttt{[CLS]}$ is denoted as $\mathbf{h}_{\texttt{[CLS]}}^l$ (i.e., $\mathbf{h}_{\texttt{[CLS]}}^l = h_1^l$). Here, the vector $\mathbf{h}_{\texttt{[CLS]}}^l$ can be regarded as the initial contextual representation for the prediction of the output of the input sequence. Subsequently, to enhance the language understanding capability, we apply a language feature transformation module composed of a linear layer, and the $\tanh$ activation function is added on top of the aggregated representation of the output, resulting in: $\mathbf{f}_i = \tanh(\mathbf{W_1}\mathbf{h}_{\texttt{[CLS]}}^l + \mathbf{b_1})$, where $\mathbf{W}$ is the weight matrix, $\mathbf{b}$ the bias term and denotes the hidden dimension size. Finally, to complete the classification task, a linear classifier is added to the feature $\mathbf{f}_i$ to obtain the output of the category distribution: $\mathbf{T}(\mathbf{f}_i, \theta_t) = \mathbf{W_2}\mathbf{f}_i + \mathbf{b_2}$. where $\mathbf{T}(\mathbf{f}_i, \theta_t) \in \mathbb{R}^K$, $\theta_t$ represent all parameters of the teacher model and $K$ represents the number of categories. Given a labelled dataset $X^l = \{x_1^l, ...x_N^l\}$ containing $N$ labelled training instances, the parameters of the teacher model will be trained by minimizing the standard cross-entropy loss function, as follows:

$$\mathcal{L}_{\mathrm{CE}}^T = -\frac{1}{N}\sum_{i=1}^{N} y_i^l \log \mathbf{T}(X_i^l), \qquad (1)$$

where $\mathbf{T}(X_i^l)$ represents the prediction distribution of the teacher model for the labelled samples, while $X_i^l$ is the one-hot encoded form of the true label. **In addition**, the teacher model is also trained through masked-language-modelling tasks to enhance its multilingual comprehension ability: $\mathcal{L}_{\text{MLM}} = -\sum_{m \in M} \log P(x_m | x_{M-m}; \theta_T)$, where $x$ is the input sequence, $M$ is all input positions, $m$ is the masked positions, and $\theta_T$ is the parameters of the teacher model and the encoder.

### 3.3.2 STUDENT MODEL

The student model is similar to the teacher model except is structurally lighter and focuses on the downstream classification task. For the input sentence $\mathbf{x}_i = [x_{i,1}, x_{i,2}, ...]$, it is obtained through the pooled output corresponding to the [CLS] token of the mBERT encoder, and is denoted as: $\mathbf{h}_i = \text{mBERT}_{\theta_s}(x_i)$. To highlight the classification features, the student model adds a nonlinear dimensionality reduction transformation module to the aggregated representation, thereby obtaining the classification feature representation: $\mathbf{f}_i = \text{ReLU}(\mathbf{W}_1 \mathbf{h}_i + \mathbf{b}_1)$. Subsequently, the prediction results' logits are generated through a linear classifier: $\mathbf{S}(\mathbf{f_i}, \theta_\mathbf{s}) = (\mathbf{W_2} \mathbf{f_i} + \mathbf{b}_2)$, where $\mathbf{S}(\mathbf{f_i}, \theta_\mathbf{s}) \in \mathbb{R}^K$, $\theta_\mathbf{s}$ is the parameter set of the student model, and K is the number of classification categories. With this foundation established, we now leverage the teacher's multilingual understanding to generate pseudo-labels for unlabelled data.

### 3.4 STEP B: META CORRECTION AND MODEL OPTIMIZATION

In the framework of teacher-student collaborative training, the quality of pseudo labels generated by the teacher's model directly affects the performance optimization of the student's model. Although the pre-trained teacher model can provide pseudo supervision signals for unlabelled data, its prediction results inevitably contain noise. When these noisy pseudo labels dominate the unlabelled data, it may lead to worse results of the student's model compared to the teacher's model. To address the problem of significant noise in pseudo labels and large differences in cross-language feature distributions in low-resource language scenarios, we introduce a multilingual pseudo label corrector. Through the meta-learning mechanism, it realizes noise perception and pseudo label optimization, further improving the reliability of semi-supervised learning. This module is based on the teacher-student collaborative framework and achieves noise filtering and cross-language feature alignment through the following mechanisms.

### 3.4.1 THE LOSS FUNCTION OF THE STUDENT MODEL

We designed a knowledge distillation loss function to optimize the training process of the student model. This loss function measures the difference between the output distributions of the student model and the teacher model using the KL divergence, encouraging the student model to simultaneously learn the pseudo soft labels $\mathbf{T}(y^u)$ generated by the teacher model as well as the probability distribution patterns $\mathbf{T}(x^u)$ output by the teacher model. Specifically, for unlabelled data $x^u$, the knowledge distillation loss is defined as: $\mathcal{L}_{\text{KD}}^u = D_{\text{KL}}(\text{Softmax}\,\mathbf{S}(x^u)/\tau || \text{Softmax}\,\mathbf{T}(x^u)/\tau)$. Here, $\mathbf{S}(x^u)$ and $\mathbf{T}(x^u)$ represent the prediction output distributions of the student model and the teacher model for the input $x$, respectively. $D_{\text{KL}}$ is the KL divergence between the two variables. $\tau$ is temperature. By minimizing $\mathcal{L}_{\text{KD}}^u$, we can effectively promote the alignment of the output distributions of the student model and the teacher model, thus improving the generalization performance of the model. Cross-entropy loss. For pseudo-label data $\mathbf{y}^u$, the probability distribution generated by the teacher model can be converted into hard pseudo-labels (one-hot vectors) through the hard Softmax operation. Then, we can learn a multilingual pseudo-label corrector to correct the potentially biased probability distribution output of the student model, thereby improving the performance of the student model. The loss of the student is as follows:

$$\mathcal{L}_{\text{CE}}^S(\mathbf{S}(x^u), y^u) = -\frac{1}{N} \sum_{i=1}^{N} y^l \log P(\mathbf{S}_i^u), \tag{2}$$

where $\mathcal{L}_{\text{CE}}^S$ represents the cross-entropy loss between the predicted labels and the true labels of the student model. $\mathbf{S}(x^u)$ represents the output distribution of the student model for the pseudo-labelled data $\mathbf{x^u}$. $\mathbf{y^u}$ indicates the pseudo-labels of the teacher model.

### 3.4.2 MULTILINGUAL LABEL CORRECTION FRAMEWORK

To enhance the reliability of pseudo-labels for unlabelled data, we have designed and introduced a multilingual pseudo-label feature corrector. By constructing a meta-learning framework to optimize the parameters of the corrector, we achieve effective correction of multilingual pseudo-label noise. We first establish a meta-learning framework, setting the meta-learning task as learning noise correction strategies from the multilingual data distribution to improve the model's generalization ability to different language noise. The support set is defined as a set of unlabelled samples in multiple languages $S = ((x_i^u, y_i^u))_{i=1}^N$, and the query set is the set of labelled samples $Q = ((x_j^l, y_j^l))_{j=1}^M$. Here, $x_i^u$ and $x_j^l$ represents the input sequence of the $i$-th unlabelled sample and the $j$-th labelled sample, respectively, and $y_i^l$ is its one-hot label vector. The goal of the meta-learner is to quickly adapt to the noise distribution in the query set using a limited support set. We adopt a meta-learning method based on double-layer gradient optimization to construct a meta-learning framework with the generalization loss of the student model on labelled data as the optimization objective. The meta-learning optimization objective: $\mathcal{L}_{\text{meta}} = E_{D_{\text{task}} \sim p(T)}[L_{\theta'}(f_{\theta'}(D_{\text{task}}^Q))]$, where $\mathcal{D}_{\text{task}} = (S, Q)$ represents the meta-learning task, $p(T)$ is the multilingual task distribution, $\theta$ is the initial parameters of the meta-learner, $\theta'$ is the updated parameters through meta-learning, $f_{\theta'}$ is the multilingual pseudo-label corrector based on the parameters $\theta'$. $L_{\theta'}$ is the noise correction loss function.

The optimization of the multilingual pseudo-label corrector is divided into two stages: the basic correction stage and the meta-learning adaptive stage. The basic correction stage uses the teacher model T to generate initial pseudo-labels $y_j^u = T(x_j^u)$ for the unlabelled samples, as well as deep feature representations $f_T$. The corrector $C_\theta$ takes these two as inputs and outputs the corrected pseudo-label distribution and confidence scores: $\bar{y}, \alpha_{conf} = C_\theta(f_T, y_j^u)$, where $\alpha_{conf} \in [0, 1]$, $\bar{y}$ represents the corrected pseudo-label, and $\alpha_{conf}$ is the confidence score.

The dual-branch input of the pseudo-label and deep features is processed by a multi-layer perceptron (MLP), and the probability of pseudo-label noise is: $P(noise_j|h_j) = \sigma(\text{MLP}(h_j))$, where $h_j = \emptyset(f_T, y_j^u)$, $\sigma$ represents the Sigmoid function, $\emptyset()$ is the feature fusion function, and $noise_j$ represents the noise indicator variable of the pseudo-label of the $j$-th unlabelled sample (1 indicates noise, 0 indicates reliable). The pseudo-labels are corrected based on the noise probability: $y_j^u = (1 - P(noise_j|h_j))y_j^u$.

### 3.4.3 META-LEARNING ADAPTIVE STAGE

By optimizing the parameters $\theta$ of the pseudo-label corrector through the meta-learning algorithm, it enables the pseudo-label corrector to quickly adapt to the noise distribution differences in multi-language scenarios. For each meta-learning task $D_{\text{task}}$, first, the pseudo-label corrector is updated rapidly using the support set S to achieve rapid adaptation and obtain the current task's parameters $\theta' = \theta - \alpha\nabla_\theta L_{\text{CE}}(C_\theta(S), y^u), S = \{(x_i^u, y_i^u)\}$, where $\alpha$ represents the meta-learning learning rate, and $L_{\text{CE}}$ is the cross-entropy loss after applying temperature scaling. Then, the meta-learning gradient is updated. The updated parameters $\theta'$ are used to correct the noise of the query set $Q = (x_j^l, y_j^l)$, and the meta-learning loss is calculated to update the initial parameter $\theta$: $\nabla_\theta L_{\text{meta}} = \beta\nabla_\theta L_{\text{CE}}(\mathbf{S}(x_j^l; \theta'), y_i^u)$. **The final loss of the student model is**: $L_S = \lambda_1 L_{\text{CE}}(\mathbf{S}(x_i^l), y_i^l) + \lambda_2 L_{\text{CE}}\mathbf{S}((x_i^u), y_i^u)$, where $\lambda_1/\lambda_2$ is the loss weight for the labelled/unlabelled data, respectively.

## 3.5 STEP C: FEEDBACK-DRIVEN UPDATE VIA BI-LEVEL OPTIMIZATION

The core objective of semi-supervised learning based on pseudo-labels within the teacher-student framework is to predict reliable pseudo-labels for unlabelled samples. In most existing studies, each training iteration typically follows a two-stage approach: first, the parameters of the teacher model are frozen to generate pseudo-labels, and then the pseudo-labels are fixed to train the student model. This one-way knowledge transfer method has significant drawbacks: the quality of pseudo-labels determine the performance upper-bound of the student model, and the error accumulation effect will cause the performance gap between the teacher and student models to gradually widen. To address these limitations and enhance the performance of the student model, we propose a feedback-driven bi-level optimization method. This approach moves beyond one-way knowledge transfer by leveraging the student's performance on labelled data as a feedback signal to directly guide the

teacher model in generating higher-quality pseudo-labels. Specifically, we formulate a collaborative evolution loop—"pseudo-label generation — student model optimization — performance feedback — teacher model update"—as a bi-level optimization problem,. We formalize this process as a two-layer optimization problem: the Inner Loop optimizes the parameters of the student model $\theta_S$ given the pseudo labels generated by the current teacher model $\theta_T$; The Outer Loop problem optimizes the parameters $\theta$ of the teacher model to minimize the loss of the student model on a clean validation set (i.e., a small amount of labelled data $D^l$), mathematically formalized as follows:

$$\min_{\theta_T} \mathcal{L}_{\text{val}}^S(\theta_S^*(\theta_T)), \quad \text{s.t.} \quad \theta_S^*(\theta_T) = \arg\min_{\theta_S} \mathcal{L}_{\text{train}}^S(\theta_S, \theta_T), \qquad (3)$$

where $\mathcal{L}_{\text{val}}^S = \mathcal{L}_{\text{CE}}^S(S(x^l; \theta_S), y^l)$ is the loss of the student model on the annotated validation set, which serves as the objective of the outer layer optimization; $\mathcal{L}_{\text{train}}^S$ is the total training loss of the student model, including labelled data and pseudo labelled data generated by the teacher (see formula (9)); $\theta_S^*(\theta_T)$ is the optimal solution to the inner layer problem, representing the optimal student parameters under the current teacher model. Due to the huge computational cost of accurately solving $\theta_S^*(\theta_T)$, we use unrolled optimization for approximation: Inner Approximation: Starting from the current student parameter $\theta_S^{(0)}$, perform K-step gradient descent to approximate the inner optimal solution:

$$\theta_S^{(k)} = \theta_S^{(k-1)} - \eta_S \nabla_{\theta_S} \mathcal{L}_{\text{train}}^S(\theta_S^{(k-1)}, \theta_T), \quad k = 1, \ldots, K \qquad (4)$$

Let the approximate solution be $\theta_S^{(K)}$, $\eta_S$ represents the learning rate of the student model. Outer Update: Using the chain rule, calculate the gradient of the outer loss on the teacher parameter $\mathcal{L}_{\text{val}}^S(\theta_S^{(K)})$ and update the teacher model:

$$\nabla_{\theta_T}^{(\text{meta})} = \frac{\partial \mathcal{L}_{\text{val}}^S(\theta_S^{(K)})}{\partial \theta_T} = \frac{\partial \mathcal{L}_{\text{val}}^S}{\partial \theta_S^{(K)}} \cdot \left(\frac{\partial \theta_S^{(K)}}{\partial \theta_T}\right), \quad \theta_T \leftarrow \theta_T - \eta_T \nabla_{\theta_T}^{(\text{meta})}, \qquad (5)$$

Where $\left(\frac{\partial \theta_S^{(K)}}{\partial \theta_T}\right)$ needs to be calculated by back-propagation along the K-step inner layer optimization path (through time back-propagation, BPTT), which allows the teacher model to perceive how its pseudo labels affect the final performance of the student model. To ensure the stability of training, the outer learning rate $\eta_T$ of the teacher model is usually much smaller than the inner learning rate $\eta_S$. **Algorithm** B provides a detailed description of the overall process, including the collaboration between Step B and Step C.

## 4 EXPERIMENTAL SETTINGS

**Dataset**: Due to the lack of MLTC datasets under a unified classification standard in real-world scenarios, we collect data from public news platforms such as Global Times and VnExpress. We use the classification tags provided by the website and classify the data into 9 categories based on the same classification criteria. The categories are: 1) Agriculture, forestry, animal husbandry and fishery; 2) Mining; 3) Manufacturing; 4) Real estate; 5) Electricity, heat, gas and water production and supply; 6) Transportation, warehousing and postal; 7) Financial; 8) Education; 9) Medical. The average word length of Chinese/English/Vietnamese/Burmese news is around 256/337/514/586, respectively. To evaluate the effectiveness of DSML, we randomly selected labelled and unlabelled data from the data. The unlabelled data are for all languages. The train/dev/test data for Chinese is 1,859/772/714, for Vietnamese is 1,266/617/604, for English is 907/527/578, for Burmese is 962/430/525 respectively.

**Evaluation**: Following previous work (Zhang et al., 2022; Chen et al., 2023; Li et al., 2024), we use **Precision**, **Recall**, and **F1** score to measure the MLTC performance. Perplexity (PPL) is used to evaluate the language comprehension ability. They are all calculated with weighted averages. More experimental settings are in Appendix A.

Table 1: Classification performance (averaged over three runs). The UDA is the base model to do the significant test for our models, all results have a statistically significant with $p < 0.01$.

| Methods | EN | | | | ZH | | | | VI | | | | MY | | | |
|---|---|---|---|---|---|---|---|---|---|---|---|---|---|---|---|---|
| | Acc | Pre | Rec | F1 | Acc | Pre | Rec | F1 | Acc | Pre | Rec | F1 | Acc | Pre | Rec | F1 |
| mBERT | 78.73 | 74.10 | 78.73 | 75.44 | 87.06 | 85.62 | 87.06 | 86.16 | 75.04 | 71.76 | 75.04 | 68.65 | 72.15 | 70.83 | 72.15 | 71.25 |
| MAML | 73.61 | 74.11 | 76.67 | 75.95 | 87.74 | 79.08 | 80.22 | 78.38 | 83.09 | 78.29 | 78.96 | 76.89 | 74.82 | 73.45 | 74.82 | 73.68 |
| UDA | 68.50 | 68.34 | 67.69 | 69.77 | 82.52 | 80.37 | 75.45 | 78.20 | 76.82 | 74.63 | 70.21 | 72.45 | 71.36 | 70.18 | 71.36 | 70.27 |
| MixText | 83.54 | 83.06 | 83.73 | 83.24 | 73.52 | 72.53 | 72.92 | 70.55 | 67.08 | 66.19 | 68.42 | 66.51 | 61.45 | 62.82 | 61.41 | 60.47 |
| JointMaTch | 87.53 | 87.45 | 87.53 | 87.38 | 81.47 | 71.14 | 80.68 | 73.28 | 74.35 | 74.18 | 77.62 | 73.91 | 69.31 | 70.19 | 69.24 | 66.25 |
| DML | 58.74 | 56.67 | 57.82 | 55.29 | 55.76 | 54.14 | 56.15 | 53.56 | 47.15 | 48.68 | 47.15 | 46.50 | 49.46 | 49.73 | 48.87 | 47.75 |
| Sailor2-8B | 79.30 | 78.82 | 79.21 | 79.14 | 79.87 | 81.35 | 79.86 | 80.60 | 78.74 | 80.21 | 78.74 | 80.18 | 76.92 | 77.85 | 76.92 | 77.24 |
| Ours | **88.15** | **88.60** | **88.10** | **88.25** | **96.26** | **96.22** | **96.26** | **96.21** | **93.28** | **93.27** | **93.28** | **93.26** | **87.03** | **87.01** | **87.03** | **86.38** |
| -PMS | 87.92 | 87.88 | 87.92 | 87.90 | 95.72 | 95.68 | 95.72 | 95.70 | 91.78 | 91.55 | 91.77 | 91.56 | 86.91 | 85.51 | 85.81 | 85.06 |
| -MNC | 88.48 | 88.43 | 88.48 | 87.61 | 95.98 | 95.97 | 95.98 | 95.96 | 92.08 | 92.07 | 92.08 | 92.03 | 85.80 | 86.25 | 86.32 | 85.63 |
| -MPC$_\&$MNC | 87.68 | 87.64 | 87.68 | 87.36 | 93.14 | 95.12 | 95.14 | 95.12 | 91.26 | 91.25 | 91.26 | 91.23 | 85.01 | 84.84 | 85.53 | 84.44 |

## 5 EXPERIMENTAL RESULTS

### 5.1 MODEL CLASSIFICATION PERFORMANCE EXPERIMENT

Experimental results are shown in Table 1, there are significant performance differences among the various baseline methods across the Chinese, Vietnamese, English and Burmese datasets. In contrast, the DSML method demonstrates strong multilingual adaptability, consistently outperforming the baselines in all four languages: Chinese, English, Vietnamese and Burmese. In Chinese, DSML achieves 96.26% accuracy and 96.21% F1, outperforming UDA by 13.74% and MAML by 8.52% in accuracy. In English, it attains 88.15% accuracy and 88.25% F1, significantly exceeding strong baselines such as MixText (83.24% F1) and JointMatch (87.38% F1). Notably, mBERT shows a marked drop in F1 from 86.16% (Chinese) to 75.44% (English), underscoring the difficulty of fine-tuning under low-resource conditions. For Vietnamese (a language with limited resources), DSML maintained the highest performance with an accuracy of 93.28% and F1 of 93.26%, while MAML decreased from 78.38% F1 (Chinese) to 76.89% F1, consistent with meta learning's reliance on richer data. In Burmese, this is a particularly challenging environment, with DSML still leading with an accuracy of 87.03% and an F1 score of 86.38%, while methods such as MixText and Joint-Match have F1 scores below 70%, highlighting our strong adaptability in situations with very low resources.

Even when compared to the large language model Sailor2-8B—which shows competitive results, especially in English (79.14% F1)—DSML achieves substantial gains: +15.61 F1 in Chinese, +13.08 in Vietnamese, and +9.11 in English. These improvements validate the efficacy of our method's design, particularly under data scarcity.

In general, the superior performance of DSML in languages - especially in low-resource settings like Vietnamese and Burmese - demonstrates its robustness and better alignment with the demands of multilinguals in the real world.

### 5.2 ABLATION STUDY

Since the main difference between DSML and the traditional dual meta-learning is: Multi-lingual Pseudo-label Corrector in Step B and Reverse Gradient Update in Step C of Figure 1, we conduct ablation study by removing these two key components. The results in Table 1 show that removing PMS causes F1 drops of 0.51% (Chinese), 1.7% (Vietnamese), 0.35% (English), and 1.32% (Burmese), confirming its role in improving pseudo-label quality. Removing MNC leads to F1 decreases of 1.23% (Vietnamese) and 0.75% (Burmese), verifying its noise correction capability. Notably, removing both modules causes larger performance degradation (e.g., 1.09% drop in Chinese), demonstrating their synergistic effect. The English dataset shows a slight F1 increase (0.06%) when removing MNC, likely due to overfitting in noise correction on smaller data, while Burmese suffers the largest drop (1.94%) when both modules are removed, highlighting their critical role for lower-resource languages.

Table 2: PPL performance with different models.

| Methods | EN | | | ZH | | | VI | | | MY | | |
|---|---|---|---|---|---|---|---|---|---|---|---|---|
| | Before | After | Final | Before | After | Final | Before | After | Final | | | |
| mBERT | – | – | 8.88 | – | – | 6.76 | – | – | 7.58 | – | – | 8.45 |
| Sailor2-8B | – | – | 9.89 | – | – | 12.94 | – | – | 7.74 | – | – | 7.86 |
| Ours | 8.52 | 8.42 | 8.47 | 6.34 | 6.30 | 6.28 | 7.97 | 7.73 | 7.57 | 7.66 | 5.51 | 7.47 |

Table 3: Performance comparison of different encoders for DSML.

| Model | EN | | | ZH | | | VI | | | MY | | |
|---|---|---|---|---|---|---|---|---|---|---|---|---|
| | Acc | Pre/Rec/F1 | PPL | Acc | Pre/Rec/F1 | PPL | Acc | Pre/Rec/F1 | PPL | Acc | Pre/Rec/F1 | PPL |
| mT5 | 88.36 | 88.16/88.36/87.53 | 8.82 | 95.92 | 95.84/95.92/95.88 | 6.24 | 92.73 | 92.77/92.73/92.72 | 7.84 | 86.54 | 86.55/86.57/85.88 | 7.98 |
| XLM-R-base | 85.42 | 85.22/84.42/84.37 | 5.05 | 93.21 | 93.22/93.21.93.16 | 7.83 | 90.51 | 90.52/90.51/90.52 | 3.48 | 83.44 | 83.43/83.34/82.74 | 3.20 |
| XLM-R-large | 89.72 | 90.22/89.72/89.67 | 4.92 | 96.11 | 96.12/96.11/96.06 | 7.77 | 93.71 | 93.72/93.71/93.70 | 3.41 | 86.54 | 86.63/86.54/85.84 | 3.14 |
| mBert | 88.15 | 88.60/88.10/88.25 | 8.47 | 96.26 | 96.22/96.26/96.21 | 6.28 | 93.28 | 93.27/93.28/93.26 | 7.57 | 87.03 | 87.01/87.03/86.38 | 7.47 |

## 5.3 MODEL LANGUAGE ABILITY EXPERIMENT

To further evaluate the language understanding capability of the teacher model, we use perplexity (PPL) as the metric. As shown in Table 2, "Before" and "After" refer to the PPL before and after fine-tuning, while "Final" denotes the overall performance. Our teacher model exhibits strong and consistent results across all languages. For English (EN), the final PPL is 8.47, showing stability from before (8.52) to after (8.42) fine-tuning. In Chinese (ZH), the final PPL reaches 6.28, with a slight decrease from 6.34 to 6.30 after fine-tuning. For Vietnamese (VI), the model achieves a final PPL of 7.57, improving clearly from 7.97 to 7.73. Notably, in Burmese (MY), fine-tuning brings a sharp PPL reduction from 7.66 to 5.51, leading to a final score of 7.47, highlighting its efficacy in low-resource scenarios. Compared to mBERT and Sailor2-8B, which only report final PPLs, our model not only performs competitively but also shows consistent refinement through fine-tuning. For example, in Chinese, our result (6.28) is better than mBERT (6.76) and far superior to Sailor2-8B (12.94). Similar advantages are observed in other languages. These results demonstrate that the teacher-student co-evolution mechanism helps the model learn robust multilingual representations. The PPL decrease after fine-tuning, especially in Vietnamese and Burmese, confirms the effectiveness of fine-tuning in strengthening the teacher model's language understanding ability.

## 5.4 EXPERIMENTS WITH DIFFERENT ENCODERS

To evaluate the impact of using different multilingual pre-trained models as the backbone for DSML, we include the larger-parameter mT5-base model (Xue et al., 2021) and extend the evaluation to XLM-R-base(Conneau et al., 2020) and XLM-R-large(Conneau et al., 2020). All models are trained on the collected dataset with consistent settings. The results in Table 3 show that our proposed teacher-student collaborative evolution mechanism remains robust across models and languages, with minimal performance differences. For both our method and models like mT5 and XLM-R, the scores of all evaluated models remain within ±0.5% of mBERT's performance. This suggests: (1) Increasing parameters does not necessarily improve performance, as representation capacity gains diminish beyond a certain scale; (2) Our mechanism effectively extracts shared knowledge across models and enhances adaptability to architectural variations through dynamic weighting and gradient fusion. In low-resource language settings including Burmese, our method consistently outperforms all baselines, demonstrating effective knowledge transfer.

## 6 CONCLUSION

This paper presents dual semi-supervised meta-learning method (DSML) for multilingual text classification. DSML uses a teacher-student co-evolution framework and contains three steps: 1) the model's foundational capabilities is trained with a small amount of labelled data; 2) the student model's multilingual classification ability is prompted with a multilingual pseudo-label corrector and the teacher model; 3) the teacher model is updated with the reverse gradient from the student until the student model is converged. Experimental results demonstrate superior performance of DSML over baselines in MLTC tasks. Meanwhile, the co-evolution mechanism also helps the teacher model to obtain a better multi-lingual comprehension ability. Further experiments on more languages are now conducting and will be released in the next version of the paper.

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

## A MORE EXPERIMENTAL SETTINGS

### A.1 DATASET ANNOTATION

To ensure the quality and consistency of the annotations, each data was independently annotated by three individuals. The final label is determined by a majority vote. For a small set (less than 50) with no majority label, we manually decide them. Then, we conduct spot checks on the majority of annotation data for all languages to resolve any ambiguity and ensure compliance with classification standards. This process ensures high-quality labels for both the initial label set and the development and testing sets. After dividing the test set, we conducted a second manual check to confirm the accuracy of the annotations in the test set.

### A.2 BASELINE MODELS

We selected existing methods based on meta-learning and semi-supervised learning as the baseline models: **mBERT** (Devlin et al., 2019) uses the averaging pooling over all the token representations from the last hidden layer as the text feature, and then inputs it into the linear classifier for prediction. We also retain the MLM head from the pre-training stage for potential language modelling tasks. **MAML** (Finn et al., 2017) optimizes the initial parameters of the model through meta-learning, enabling the model to quickly adapt to new tasks with only a small number of samples and a few gradient updates. **UDA** adds random perturbations (such as noise, word embedding masks) to the input samples, requiring the model to produce consistent prediction results for the original samples and the perturbed samples. **MixText** (Chen et al., 2020) employs a novel hidden-space interpolation technique (TMix) for data augmentation to effectively leverage unlabeled data in semi-supervised text classification, particularly under limited supervision.**JointMatch** (Zou & Caragea, 2023) introduces adaptive classwise thresholds and a cross-labeling mutual learning scheme between two

networks to mitigate pseudo-label bias and error accumulation in semi-supervised text classification. **DML** (Li et al., 2024) combines the dual learning framework with semi-supervised learning to improve the quality of pseudo-labels in semi-supervised text classification, while also optimizing the teacher and student classifiers. **Sailor2-8B** (https://huggingface.co/sailor2) is a large language model designed for multilingual scenarios in Southeast Asia, supporting 15 languages.

### A.3 PARAMETER SETTINGS

We employ the multilingual BERT (mBERT) model as our encoder. The hyper-parameters are optimized based on performance on the validation set. The maximum input sequence length is set to 128, with a batch size of 16. To mitigate overfitting, we apply a dropout rate of 0.1. The model is optimized using AdamW with a learning rate of 5e-5 and a weight decay of 0.01.For the knowledge distillation framework, we set the teacher and student temperatures to 2.0 for the Softmax function. The distillation loss weight $\alpha$ is set to 0.5, balancing the contribution between the original task loss and the distillation objective. Additionally, we introduce a noise ratio of 0.1 for data augmentation to enhance model robustness. A confidence threshold of 0.7 is empirically selected for pseudo-label selection. All experiments are conducted on an NVIDIA RTX 4090 GPU.

## B ALGORITHM DEMONSTRATION FOR DSML

---

**Algorithm 1** DSML: Co-training Algorithm based on Bi-level Optimization

---

**Require:** Labelled data $D^l$, unlabelled data $D^u$, inner steps $K$, learning rates $\eta_S, \eta_T$
**Ensure:** Optimized teacher parameters $\theta_T$ and student parameters $\theta_S$
1: **Initialize** teacher parameters $\theta_T$ and student parameters $\theta_S$ (via Step A)
2: **while** student model not converged on $D^l$ **do**
3:     **Step B: Meta-correction and Model Optimization**
4:     Teacher generates pseudo-labels: $y^u = T(x^u; \theta_T)$
5:     Multilingual pseudo-label correction: $\tilde{y}^u, \alpha = C_m(y^u, f_T)$
6:     Student computes loss $\mathcal{L}_{\text{train}}^S$ using $\tilde{y}^u$ and $D^l$ (Eq. (3),(9))
7:     **Step C: Feedback-driven Bi-level Optimization**
8:     // Inner loop: Optimize student model
9:     $\theta_S^{(0)} \leftarrow \theta_S$ {Save initial state}
10:     **for** $k = 1$ **to** $K$ **do**
11:         $\theta_S^{(k)} \leftarrow \theta_S^{(k-1)} - \eta_S \nabla_{\theta_S} \mathcal{L}_{\text{train}}^S(\theta_S^{(k-1)}, \theta_T)$
12:     **end for**
13:     $\theta_S \leftarrow \theta_S^{(K)}$ {Update student model}
14:     // Outer loop: Optimize teacher model
15:     Compute validation loss: $\mathcal{L}_{\text{val}}^S = \mathcal{L}_{\text{CE}}^S(S(D^l; \theta_S), y^l)$
16:     Compute meta-gradient via BPTT: $\nabla_{\theta_T}^{(\text{meta})} = \frac{\partial \mathcal{L}_{\text{val}}^S}{\partial \theta_T}$ {Backpropagate through K steps}
17:     Update teacher: $\theta_T \leftarrow \theta_T - \eta_T \nabla_{\theta_T}^{(\text{meta})}$
18: **end while**

---

## C ERROR ANALYSIS

To thoroughly evaluate the performance of the DSML model and understand its error patterns, we conducted a detailed analysis that combined quantitative metrics with qualitative case studies.

We began by examining the model's confusion matrix on the test set (Figure 2). The matrix reveals that while the model performs well overall, certain systematic errors exist. For example, a number of samples from the *Mining* category (Label 2) are misclassified as *Manufacturing* (Label 3). This suggests that the model struggles to distinguish between these two semantically related categories, which often share contextual keywords related to production and output.

This pattern is exemplified by the following case:

*Text: "On April 25th, Norilsk Nickel of Russia announced on Thursday that its consolidated nickel output in the first quarter increased by 3.4% compared with the same period last year to 56,000 tons. The company said that the consolidated copper output in the first quarter was 127,000 tons, an increase of 13% year-on-year."*
**True Label:** 2 (Mining)
**Predicted Label:** 3 (Manufacturing)

We hypothesize that the model is misled by high-frequency words such as "output," "consolidated," and "company," which are strongly associated with manufacturing contexts. Consequently, it overlooks the core thematic cue—the extraction of raw materials like nickel and copper—which defines the mining industry.

In contrast, mining texts that contain more distinctive verbs (e.g., "excavate," "extract") or nouns (e.g., "mine," "pit") are typically classified correctly. This contrast highlights that the model's performance is highly sensitive to the specific terminology present and may rely on superficial lexical cues rather than a deeper understanding of the core activity.

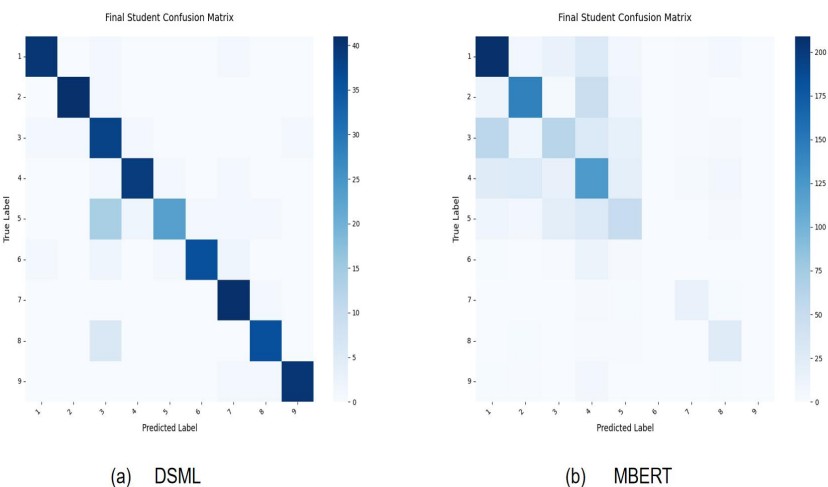

(a)  DSML

(b)  MBERT

Figure 2: Confusion matrix of the DSML model on the test set

# D  EXPERIMENTAL SETUP FOR LARGE LANGUAGE MODEL

To ensure fair comparison with the baseline of Large Language Models (LLM), we evaluated the Sailor2-8B model using the Zero shot prompting method; Tip Template: We have designed a structured tip template to convert text classification tasks into a format that LLM can easily understand. The template includes task description, category list, and text to be classified. The decoding strategy uses greedy decoding to ensure output certainty and reproducibility; Post process the raw output of LLM. We first extract the first complete word or phrase from the generated text, and then match it precisely with a predefined list of categories to determine the final predicted category. The prompt template is as follows: (Taking English as an example):

Prompt:

You are a text classification expert. Classify the following news article into one of these categories: [Agriculture, Mining, Manufacturing, Real Estate, Power Utility, Transportation, Finance, Education, Healthcare].

Article: {input_text}

Category:

Figure 3: Prompt template for large language model

