# OpenReview forum: "DSML: Dual Semi-Supervised Meta-Learning for Multi-Lingual Text Classification"
_ICLR.cc/2026/Conference — Submitted to ICLR 2026_

### Official Review · Reviewer_KoPH · 2025-10-29

**Soundness:** 3
**Presentation:** 2
**Contribution:** 3
**Rating:** 6
**Confidence:** 4

**Summary:**

The paper introduces DSML, a dual semi-supervised meta-learning framework for multilingual text classification under low-resource conditions. It builds a teacher–student co-evolution architecture in which both models are first initialized with a small labeled set, the teacher additionally being trained with MLM to enhance multilingual understanding. During training, the student learns from teacher-generated pseudo-labels with a multilingual pseudo-label corrector and distillation, while a bi-level meta-optimization loop feeds back the student’s performance on labeled data to update the teacher, improving pseudo-label quality over iterations. Experiments on a newly collected multilingual dataset (Chinese, Vietnamese, English, Burmese) show that DSML achieves state-of-the-art performance without relying on external knowledge, outperforming strong baselines and demonstrating robustness in low-resource languages.

**Strengths:**

Originality
- Proposes a dual semi-supervised meta-learning formulation that couples multilingual comprehension (via teacher) and classification (via student) through a bi-directional co-evolution loop, rather than the standard one-way distillation/self-training.
- Introduces a multilingual pseudo-label corrector optimized by a meta-learning loop.
- Demonstrates that semi-supervised and meta-learning ideas can be synergistically combined to address multilingual low-resource settings without external knowledge, which is an under-explored junction.

Quality
- The algorithmic pipeline is fully specified (step-wise A/B/C with explicit loss definitions, bi-level optimization argument, unrolled inner loop).
- Evaluation spans four languages including low-resource ones, which is more demanding than monolingual SSL settings.
- Ablation (-PMS, -MNC) confirms that both innovations contribute, not just scaling or incidental tuning.
- Statistical significance test (p<0.01 vs UDA) supports non-accidental gains.

Clarity
- The paper maintains a clean separation of roles: teacher for multilingual understanding; student for classification; corrector for noise mitigation; meta-loop for feedback refinement.
- Step-wise exposition (A/B/C) provides conceptual scaffolding that helps track where knowledge flows and where gradients act, reducing conceptual ambiguity.

Significance
- Addresses a real and persistent bottleneck in MLTC: scarcity of aligned labels across languages and the fragility of pseudo-labels when languages are typologically far.
- Achieves consistent SOTA gains, most notably in low-resource languages, where the community currently lacks robust baselines.
- Avoids reliance on external ontologies or translations, increasing deployability in realistic resource-constrained settings.
- Provides a credible blueprint showing that teacher–student frameworks need not remain one-directional; feedback-driven co-evolution can unlock new performance ceilings for multilingual SSL.

**Weaknesses:**

Ambiguity in the definition and reporting of key metrics (e.g., “Final” PPL)
- Before/After/Final are not rigorously defined, and the subject (teacher vs student vs joint) is unclear, reducing interpretability and reproducibility. Please precisely specify which model/state the “Final” column reflects and unify metric semantics.

Insufficient and non-aligned comparison against LLM baselines
- While the paper includes Sailor2-8B as a reference point, the comparison is not aligned in terms of supervision regime or optimization objective — it is unclear whether the LLM baseline was adapted under the same limited-label / semi-supervised constraints or merely reported “as-is.” This substantially weakens the evidential weight of the comparison, given that LLMs already embed strong multilingual priors. In addition, no experiments assess DSML under LLM-equivalent variants such as “LLM as teacher,” LLM-based pseudo-labeling, or LLM-assisted few-shot adaptation with identical annotation budget. Without cost-normalized and supervision-matched baselines, it remains unclear whether DSML offers a genuine advantage over modern multilingual LLMs or whether the reported gain is merely due to asymmetric evaluation settings.

**Questions:**

- PMS and MNC were never explicitly defined in the main text before use in Table 1. Would you clarify their full names and exact functional isolation to ensure interpretability of the ablation table?
- Dual-meta SSL and multilingual SSL baselines exist (e.g. Li et al. 2024). Can you provide direct comparisons or a sharper justification of what DSML adds beyond those formulations?
- Bi-level learning with unrolled updates can be unstable and expensive. How many unrolled steps were used? What is the actual compute budget relative to baselines? Are the reported gains robust across random seeds?
- The results are based solely on a newly collected dataset. How were labels aligned across languages to ensure semantic equivalence?Was inter-annotator agreement measured? Will the dataset be released, or can you include at least one public benchmark (MLDoc or similar) to validate generalizability?

---

### Official Review · Reviewer_pPmH · 2025-11-01

**Soundness:** 3
**Presentation:** 2
**Contribution:** 2
**Rating:** 2
**Confidence:** 4

**Summary:**

This paper proposes a method called Dual Semi-Supervised Meta-learning for enhancing the model's multilingual text classification capability. Unlike previous approaches, in this method, both the teacher model and student model update their parameters simultaneously to overcome the limitations of the teacher model's performance and achieve superior results. The method demonstrates effectiveness across multiple base models, with the paper reporting state-of-the-art results on a new benchmark.

**Strengths:**

The teacher model also updates parameters in the proposed methodology, and the final results after training on mbert surpass existing approaches.

**Weaknesses:**

The article contains several several sections lacking sufficient detail, making it difficult to understand. The method was tested only on one type of task, and some experimental designs are not sufficiently justified.

**Questions:**

1, line 19-20: granularity->gradually
2, What is PMS, MNC and MPC in Table 1 and Sec 5.2? Please give clear definitions, e.g., Multi-lingual Pseudo-label Corrector (MPC), Reverse Gradient Update (RGU).
3, In Table 2, does the final score of mbert represent the performace of the original mbert model? What is the difference between mbert's final and Ours before, given that you use mbert as the base model?
4, In Table 3, Zh row of XLM-R-base: 93.22/93.21.93.16 -> 93.22/93.21/93.16
5, Table 3 should compare the performance of different models before and after training, to better demonstrate robustness.

---

### Official Review · Reviewer_kPyz · 2025-11-01

**Soundness:** 3
**Presentation:** 3
**Contribution:** 2
**Rating:** 4
**Confidence:** 5

**Summary:**

This paper addresses the challenges of multilingual text classification (MLTC)—namely, scarce labeled data and linguistic disparities—by proposing a Dual Semi-supervised Meta-Learning (DSML) framework. DSML employs a teacher-student architecture with a co-evolutionary mechanism to enhance both multilingual understanding and classification performance under low-resource conditions. The approach operates in three stages: (1) initializing both teacher and student models with limited labeled data, where the teacher starts with superior multilingual capability; (2) refining the student’s multilingual classification ability through pseudo-labeling guided by a multilingual pseudo-label corrector and the teacher model; and (3) evaluating student convergence using a small validation set, and if not converged, updating the teacher model via backpropagation from the student to iteratively improve pseudo-label quality. Extensive experiments on a newly curated MLTC dataset demonstrate that DSML achieves state-of-the-art performance.

**Strengths:**

1. The paper presents DSML, a semi-supervised meta-learning approach that alternates between generating pseudo-labels with a teacher model and updating both teacher and student based on validation feedback. This iterative co-update mechanism differentiates it from standard teacher-student or self-training methods.

2. The evaluation is conducted on a newly collected multilingual text classification dataset, where DSML shows consistent improvements over baselines, especially in low-resource settings. The release of data and code also supports reproducibility.

**Weaknesses:**

1. The method lacks explicit multi-lingual modeling design. Its core mechanisms (teacher–student co-evolution, pseudo-label correction, and bi-level optimization) are generic semi-supervised learning techniques that do not specifically address cross-lingual transfer, alignment, or language divergence. Consequently, DSML’s gains appear to stem primarily from effective low-resource semi-supervised learning rather than from innovations tailored to the unique challenges of multi-lingual text classification.

2. The paper lacks comparison with state-of-the-art large language models (e.g., Llama-3, Qwen-3), which are now widely used as strong multilingual baselines. Given that these models exhibit strong zero-shot or few-shot cross-lingual capabilities, their absence makes it difficult to assess whether DSML’s gains are competitive in the current LLM-dominated landscape or merely reflect improvements over older architectures like mBERT.

3. The bi-level optimization between teacher and student models incurs substantial computational. However, the paper does not report training time, memory consumption, or FLOPs, making it impossible to assess whether the performance gains justify the added complexity.

**Questions:**

None.

---

### Official Review · Reviewer_iJay · 2025-11-01

**Soundness:** 3
**Presentation:** 2
**Contribution:** 2
**Rating:** 6
**Confidence:** 3

**Summary:**

The authors propose a teacher-student meta-learning framework toward the problem of multi-label multi-languages text classification using semi-supervised learning to augment the low-level of labeled data.

**Strengths:**

- MLM-loss for teacher training using labeled data
- Building a label augmentation and meta-correction of the unlabeled data
- Separate loss functions of teacher and student models

**Weaknesses:**

- Not comparing against an LLM that is trained on a specific language
- Not studying classification of imbalanced classes

**Questions:**

- Are the results general to other groups of languages? What if the language of interest does not belong to the set of languages used to train the multilingual BERT?
- Can you include the graph of the model learning / convergence with time?

---

### Official Review · Reviewer_VjL2 · 2025-11-06

**Soundness:** 2
**Presentation:** 1
**Contribution:** 2
**Rating:** 2
**Confidence:** 4

**Summary:**

This submission proposes a method for multilingual, semisupervised text categorization, combining pseudo-labeling, pseudo-label correction, and some meta-learning. On a new dataset covering four languages (fairly well resourced), the method yields consistent performance improvements over a number of alternatives.

**Strengths:**

Multilingual text categorization is certainly a relevant problem, and particularly suited to semi-supervised learning as well.

A new multilingual TC dataset is introduced, that will hopefully become publicly available.

**Weaknesses:**

The paper greatly oversells the novelty of the proposed task and approach. There has been lots of work on multilingual text categorization, including semi-supervised and multiview learning. One example is the work of Usunier and Amini around 2010-2011, which addresses similar challenges in the pre-deep-learning era. Some sizeable datasets have been used in that context, including a freely available 5 language corpus derived from the Reuters corpus (doi 10.24432/C5FS5R) that is much larger that the proprietary corpus used here.

The data used in the paper is also disappointing. It is fairly small by current standards, especially for a task motivated by the idea that there is  massive amount of unlabelled data to leverage.

The main weakness is clarity and presentation in both the methodology and experimental sections. There is little actionable detail on the various parts of the framework, except introducing some notation (confusing, more details below) and losses. In the experimental section, there is no detail in the main paper on how the various competing techniques, and appendix A.2 does not clarify how they were used on this task (for example, mBERT and MAML are not natively semi-supervised). Re. results, some significance testing was done, which is good, but one understands significance was tested versus UDA, which is among the lowest performing techniques. Without pairwise testing between the best methods, or some assessment of the experimental uncertainty, it is difficult to know how significant observed differences are. Finally, lots of experimental results are odd, in particular, many F-scores are not between the precision and recall reported. As the (harmonic) average of p and r, F1 is by construction between p and r. Likely a side-effect of some weird averaging, but again there is no detail describing how that was done.

**Questions:**

Sec. 3.3.1: This seems to describe a standard mBERT encoder with a one-layer MLP classifier output. However a lot of the description stays high-level while the notation is confusing. What is meant by "the prediction of the output of the input" (l.204)? Isn't the output of a classifier the class predictions, essentially? Is there a f_i (l.207) for each token (i indexes token)? The encoding of the CLS token would presumably be the same for all. Eq. 1: One would expect the cross-entropy to be between the predictions T(f, \theta) and output, yet it is between the output and "labels" -- it is really hard to make sense of this portion without clarifying or straightening the notation.

Sec. 3.3.2: In what way is the student "structurally lighter" (l.224)? From the short description, it looks essentially identical, apart from using a ReLU instead of tanh. Encoding and MLP layer are identical down to the notation. (One assumes W_1, b_1, W_2, b_2, f_i, etc. are actually not the same as in Sec. 3.3.1 -- this should also be ironed out). The description mentions a "nonlinear dimensionality reduction", yet no dimensions are given (apart from the K-dim. classifier output).

l.370: Presumably these numbers are *document* lengths *in words*, rather than word length? And numbers in l.373-374 are number of documents?

l.375: These are very weird references for precision/recall/F-score.

Typos:

l.048: "multi-language" -> multilingual?

l.051: "generality ability" -> generalization?

l.073: what is meant by "jointly prompts" in this context?

l.107: dupe ref. Hatefi et al. (2025)

l.108: DML is not introduced. Same with MAML, PBML, EMPT later (but at least refs. are provided for those). Also acronym MLM (l.147).

l.157: student and teacher *are*

l.208: "the bias term and denotes" -- something missing here.

---

### Meta-Review · Area_Chair_m2dB · 2025-12-09

**Summary:**

The paper introduces a dual semi-supervised meta-learning framework for multilingual text classification under low-resource conditions. It is based on a teacher–student co-evolution architecture.  Through a co-evolution mechanism, the student model granularity learns the teacher model's multilingual comprehension ability.

**Reviewer Concerns:**

The novelty is rather limited.  There has been lots of work on multilingual text categorization, including semi-supervised and multiview learning. One example is the work of Usunier and Amini around 2010-2011, which addresses similar challenges in the pre-deep-learning era. Some sizeable datasets have been used in that context, including a freely available 5 language corpus derived from the Reuters corpus (doi 10.24432/C5FS5R) that is much larger that the proprietary corpus used here.

The data used in the paper is also disappointing.

The method lacks explicit multi-lingual modeling design.

The paper lacks comparison with state-of-the-art large language models (e.g., Llama-3, Qwen-3), which are now widely used as strong multilingual baselines.

The bi-level optimization between teacher and student models incurs substantial computational. However, the paper does not report training time, memory consumption, or FLOPs.

**Reviewer Scores:**

There are many issues of this paper, which are listed above.

But the answers to these issues are not satisfied.

---

### Decision · Program_Chairs · 2026-01-26

Reject